# Quality Assessment of Waste from Olive Oil Production and Design of Biodegradable Packaging

**DOI:** 10.3390/foods11233776

**Published:** 2022-11-23

**Authors:** Joanna Grzelczyk, Joanna Oracz, Ilona Gałązka-Czarnecka

**Affiliations:** Institute of Food Technology and Analysis, Faculty of Biotechnology and Food Sciences, Lodz University of Technology, 90-537 Lodz, Poland

**Keywords:** pomace, olives, disposable tableware, biodegradable, antioxidants, polyphenols

## Abstract

The use of olive pomace from olive oil production is still insufficient. The lingering olive pomace is harmful to the environment. On the other hand, the world is increasingly polluted with plastic or by-products from the production of biodegradable products. Considering these two problems, the aim of this work was to develop a mixture and create biodegradable disposable tableware characterized by high antioxidant activity. The disposable tableware was made by mixing olive pomace with teff flour or/and sorghum groats and lecithin. Baking was carried out at the temperature of 180 °C. The best variant of the mixture for the preparation of disposable tableware was olive pomace, teff flour, sorghum groats and lecithin. These vessels were the toughest, with low water absorption and had a high antioxidant potential due to the high content of polyphenols and omega acids. Protecting the cups and bowls with beeswax had a positive effect on reducing water absorption.

## 1. Introduction

Olive oil is mainly produced in Europe. World olive production was about 2.86 million tons from 2005 to 2018. An important fact is that the preparation of such an amount of oil requires the use of 14.3 to 17.6 million tons of olives, of which about 30% is waste in the form of pomace [1,2]. The latter is an unstable material with a high-water content [3,4]. However, it is a rich source of many valuable components, including omega-3 and -6 fatty acids, mineral compounds, vitamins, carbohydrates, polyphenols and fiber [5,6,7]. Waste generated in oil production is already successfully used for animal feed or biomass as a source for biofuels [1,8]. Nevertheless, such solutions still have not brought the expected effects. The insufficient management of olive waste is not only a local problem but also a global one. The large mass of pomace produced in a relatively short period of time is an important element for processing plants [8,9].

Moreover, it is also well known that the amount of used disposable plastic tableware continues to grow rapidly, this growth consequently leading to environmental and economic problems that affect not only most developed countries but also the entire ecosystem of the whole planet due to the improper management of such waste [10,11]. To date, disposable tableware was always made of plastic, i.e., polypropylene, polyethylene, or polyethylene terephthalate [12,13,14]. The usable advantage of such materials is their durability and stability, but on the other hand they have a very long decomposition time. Ironically, the durability of such plastic packaging and dishes, which is a desirable feature, results in their lingering in the oceans and rather than undergoing mineralization they are consumed by marine organisms [15,16,17,18,19]. Starch and bioplastics, in the Life Cycle Assessments (LCAs) process, display a reduced generation of carbon dioxide emissions into the atmosphere [20,21,22,23,24,25]. The bioplastics based on biomass sources derived from industry by-products or waste reduce the emissions of GHG, since no additional resources are implied in the raw materials production [26,27]. To reduce the content of carbon dioxide emissions, packaging production time can be shortened, but plant waste can also be used, the deposition of which, in landfills, increases the emission of carbon dioxide [28]. Therefore, new packaging materials are being extensively explored and other agricultural products such as bran and corn have been recently used [29,30]. However, these are products of daily necessity in the production of food, e.g., bread and there is increasingly insufficient food in the world, to which 1 in 9 people do not have adequate access [31]. Climate change, which causes floods or droughts, also affects the growth deficit of agricultural products [32]. For this reason, the production of new biodegradable products is highly profitable.

The aim of this research is to develop a mixture of disposable tableware made of olive oil waste and to qualitatively assess the nutritional value.

## 2. Materials and Methods

### 2.1. Chemicals and Reagents

2,2-diphenyl-1-(2,4,6-trinitrophenyl) hydrazyl (DPPH), 6-hydroxy-2,5,7,8-tetramethylchroman-2-carboxylic acid (Trolox), methanol, water and galic acid was purchased from Sigma-Aldrich (St. Louis, MO, USA). Folin–Ciocalteu’s phenol reagent, sodium carbonate (Na_2_CO_3_) and sodium nitrate were purchased from Chempur (Piekary Slaskie, Poland). Ultrapure water (resistivity 18.2 MΩ cm), obtained from a Milli-Q purification system (Millipore, Bedford, MA, USA), was used for all analyses. Teff flour, sorghum groats, lecithin, olives and beeswax were purchased at local stores.

### 2.2. Material and Preparation

Biodegradable disposable dishes (in the shape of cup, plate or bowl) were prepared from by-products of olive oil production (69.5–79.5%) and flour and/or groats and lecithin in various combinations and concentrations (Table 1). The olive pomace is made from green olives of the Manzanilla variety from Spain, cold-pressed. Olive pomace is the residue of pulp and skin, the initial moisture content of which was 51% by weight. The flour/groats weight ratio of 1:1, at 30–20% by weight and lecithin in liquid form of rapeseed origin at 0.5% by weight and flour and groats with a grain size of less than 0.5 mm were used. All ingredients were mixed to form a firm homogeneous mass which was filled into a baking tray. The stuffed disposable dishes were baked in a hot air oven with top and bottom heating for 1.5 h at 180 °C. After baking, the dishes were allowed to cool down (24 °C). Examples of biodegradable disposable utensils are shown in Figure 1.

In order to use the cups for liquid dishes, the cups were covered with beeswax. The beeswax was dissolved at a temperature of 70 °C, then the beeswax was evenly applied with a brush on the disposable dishes and heated for 2 min at a temperature of 60 °C. The dishes were then cooled to room temperature.

All packages were stored at room temperature in a dark place (in a cabinet) without additional packages.

The mixture and the method of production can be found in the National patent application 437771 (Section 5).

### 2.3. Physical and Mechanical Properties

#### 2.3.1. Flexural Strength

The flexural strength of the disposable utensils was measured instrumentally by 3-point bending test jig flexural test with the EZ Test texturometer (Shimandzu, Kyoto, Japan) using Trapezium software. The procedure was performed according to Olt et al. [29] with minor modifications. The flexural strength test was carried out by crushing a 4 × 4 × 0.3 cm piece of the studied materials, which was placed on two shaped supports with their centres positioned L = 34 mm apart and with a load subsequently being applied to the central part of the test body from the top to the bottom using a knife-shaped element. The distance between fulcrums was 10 mm, test speed 0.75 mm/min and the load was applied until the test body broke, the force necessary to achieve such a break being measured. Five measurements were made. The force sensor was automatically calibrated before the tests using the appropriate calibration cable (Calibration certificate number L1.436.3114.2022). The analyzes were performed on day 1 after baking and after one week, 2 weeks, 4 weeks and 8 weeks of storage. Five repetitions were made.

#### 2.3.2. Drop Test

Samples of disposable utensils with no cracks were dropped from a height of 0.7 m and were thereafter analyzed for cracks or splits after impacting on a level floor. The disposable utensils were then subjected to a stress test [33]. The analyzes were performed on day 1 after baking, after 4 weeks and 8 weeks of storage. Five repetitions were made.

#### 2.3.3. Measurement of Color Parameters

Color parameters were measured in the CIE L*a*b* system, using a Konica Minolta CR-400 (Tokyo, Japan) colorimeter, with SpectraMagicTM NX software, equipped with a CR-A33a protective glass measuring head. In the L*a*b* system, the a* value indicates the proportion of red, the positive b* value the proportion of yellow and L* the brightness of the sample. Measurements were made using the D65 illuminant (PN-65-N-01252), which is the average distribution of daytime radiation power at different times of the day (with UV), at different latitudes of European countries with varying degrees of cloud cover. The analyzes were performed on day 1 after baking, after one week, 2 weeks, 4 weeks and 8 weeks of storage.

#### 2.3.4. Differential Scanning Calorimetry (DSC)

Thermal properties of the composite disposable utensils were analyzed using DSC 2 METTLER TOLEDO (Columbus, OH, USA), at a heating rate of 5 °C/min. Approximately 4 mg of sample was placed in a Tzero^®^ (Columbus, OH, USA) aluminum pin hole hermetic pan, which was heated from −25 °C to 100 °C. The DSC was calibrated with indium. A high purity nitrogen was used as the purge gas with flow rate of 50 mL/min. Samples were analyzed before and after baking.

#### 2.3.5. Water Activity

The water activity in the tested samples was measured with the HP23-AW-A Rotronic (ROTRONIC Instrument Corp, Hauppauge, NY, USA) water activity meter in combination with the HC2-AW Rotronic salt probe, which enables the measurement of the water activity of a whole product. Before the measurement, 2 g of biodegradable dish was ground in a mortar and placed in a desiccator to equilibrate the humidity. The sample was then transferred to a cup and placed in the probe. The measurement was taken. The analyzes were performed on day 1 after baking, after one week, 2 weeks, 4 weeks and 8 weeks of storage.

#### 2.3.6. Water Absorption

Cups and bowls with/without beeswax were weighed empty. Then 40 mL of cold or hot water was added. The dishes were weighed again. They were left for 6 h. The analysis was stopped when the first drop appeared (water leakage) under the package [33]. The analyzes were performed on day 1 after baking, after one week, 2 weeks, 4 weeks and 8 weeks of storage.

#### 2.3.7. Biodegradability Test

The biodegradation was tested on five pieces of each disposable utensil, with 6 × 6 cm samples, of known weight, placed in ground for growing flowers. The samples were retrieved every 7 days from where they were buried under soil, dried at room temperature and placed in a desiccator. Before weighing, each sample was thoroughly cleaned of soil [34]. The analyzes were performed on day 1 after baking, after one week, 2 weeks, 4 weeks and 8 weeks of storage. Three repetitions were made.

### 2.4. Nutritional Properties

#### 2.4.1. Antioxidant Activity of the Extracts by DPPH Assay

The antioxidant activity of the extracts was measured by the DPPH assay and determined according to Zielińska et al. [35] with small modifications. The disposable utensils were ground in a mortar. Next, 1 g was placed into a 50 mL flask and 20 mL of 70% methanol was added. The samples were shaken for 1 h, then filtered. A 0.50–0.25 mL volume of the sample was mixed with 1.95–1.75 mL of a 0.50 mL solution of DPPH in 70% methanol. The samples were homogenized and incubated at room temperature for 30 min in darkness. The absorbance was measured at 517 nm and the control used was 70% methanol. Control samples were prepared in the absence of extracts, following the same procedure. The IC50 value denotes the concentration of the sample required to scavenge 50% of the DPPH free radicals. The scavenging effect was calculated according to Equation: Antioxidant activity (%) = ((AControl − AExtract)/AControl) × 100, based on the calibration curve of Trolox, whose linearity range varied between 20 and 500 μmol/L. The analyzes were performed on day 1 after baking, after one week, 2 weeks, 4 weeks and 8 weeks of storage.

#### 2.4.2. Total Phenolics Content (TPC)

The content of total phenolic compounds (TPC) of the extracts by the Folin–Ciocalteu assay was determined according to a modified procedure from Mattia et al. [36]. The disposable utensils were ground in a mortar. Next, 0.5 g was taken into a 50 mL flask and 20 mL of deionized water was added. The samples were shaken for 1 h, then filtered. An aliquot of the extracts (0.1 mL) and 4.9 mL of deionized water were mixed with 500 μL of Folin–Ciocalteu reagent and allowed to stand for 3 min. Next, 1.5 mL of a 25% Na_2_CO_3_ solution was added and then deionized water up to 10 mL final volume. After incubation in darkness for 1 h, the absorbance at 765 nm was measured. The results were expressed as gallic acid equivalents (GAE) (g/100 g) of extract using a standard curve for gallic acid, ranging from 12.5 to 1000 μg/mL. The analyzes were performed on day 1 after baking, after one week, 2 weeks, 4 weeks and 8 weeks of storage.

### 2.5. Statistical Analysis

Results were expressed as the mean value ± standard deviation (SD). Statistical tests were evaluated by using the Statistica 13.0 software (StatSoft, Inc., Tulsa, OK, USA). Analysis of variance (ANOVA) and the Tukey post-hoc tests were applied to determine differences between means. Differences were considered to be significant at *p* < 0.05.

## 3. Results

### 3.1. Mechanical and Thermal Properties

Determining the mechanical properties of food packaging such as cups, plates, or disposable bowls is essential for the achievement of high-quality products. They are mainly analyzed by compression/crush testing to determine the elasticity and the deformation and plasticity limit. Thus, the compressive strength of the product can be determined [37]. The results of the analysis of the mechanical properties of disposable utensils are presented in Table 2.

Disposable utensils produced from the mixture of olive pomace, teff flour and sorghum groats (OTS) showed the lowest crushing force of 7.45 N/mm^2^. The remaining mixtures showed a similar crushing force of about 10 N/mm^2^ due to the addition of lecithin. However, during the storage period, disposable tableware needs higher flexural strength. Disposable utensils obtained from the mixture of olive pomace, teff flour, sorghum and lecithin (OTSL) after 8 weeks had the lowest flexural strength 11.45 N/mm^2^, compared to those prepared from the mixture of olive pomace, teff flour and lecithin (OTL) and a mixture of olive pomace, sorghum and lecithin (OSL). OTS was the hardest after 8 weeks. Interestingly, the protection of disposable tableware with shin wax did not significantly affect the crushing force. Similar results were obtained by Olt et al. [29], who showed that the flexural strength of rye and wheat bran plates was 14 N/mm^2^ and that of tableware made of hulls and buckwheat bran was less than 9 N/mm^2^. In view of the above, it can be concluded that the food packaging developed in this study has adequate crushing strength, which in the case of disposable tableware should not be lower than 10 N/mm^2^ [29]. Liu et al. [37] developed cellulose-based cups using sugarcane pomace and bamboo fibers with three times greater crush strength than the products analyzed. The Liu research team also tested the crushing force of a plastic cup, which was in the polystyrene range of 15.6 N/mm^2^ [37]. Our material is less crush-resistant compared to the polystyrene cup [37]. Considering that the disposable dishes from the mixtures developed in this study need to be suitable for consumption, the lower squeezing force is an advantage because the material will be much easier to be consumed (crush in the mouth).

In the next stage of the research, the produced disposable packaging was subjected to color analysis. As shown in Table 3, there are differences in the packaging color without lecithin compared to the utensils from the mixture with lecithin. OSL, OTL and OTS packaging was brighter compared to OTSL. The a* value of OTS indicated a significant difference (*p* < 0.05) between the OTSL, OTL and OSL samples and contained a greater degree of redness. The L* value indicates the brightness of the products, which have a gray-brown color. The color did not change significantly, only slightly darkening with storage time, and not noticeable to an ordinary consumer. The addition of beeswax did not significantly change the color of the packaging, and only in large amounts. As the tested samples were immersed in the wax and then the wax surface was evened out with a brush, its thickness was so small that it did not brighten the final product. Disposable dishes were dark after baking, which also resulted in a lack of visual changes in the product. Beeswax does not contain a strong, permanent dye [38,39].

The pomace generated during the production of olive oil contains fatty acids and may oxidize over time. A common knowledge is that the enrichment of pomace with other ingredients including nutrients, can change unfavorable properties, enabling commercial utilization. For example, the addition of antioxidants can reduce the oxidation of sensitive components, while the addition of flour can change the thermal parameters of thermally unstable raw materials. The present study was conducted for packaging with the addition of lecithin, which in this case was used as an antioxidant. Differential scanning calorimetry (DSC) was used to demonstrate changes in the thermal parameters of the tested samples. The results of DSC analysis of the samples taken before and after baking were analyzed (Figure 2). The DSC curve of the unbaked samples is characterized by a single endothermic peak observed in the temperature range of 75–82 °C. The melting temperature peaks (Tm) of the unbaked OTSL, OTL and OSL samples were observed at temperatures of 77 °C, (heat capacity, 105 J/g), 79 °C (112 J/g) and 78 °C (109 J/g), respectively. This temperature distribution may be due to the diversity of components in the matrix including not only non-oxidizing fatty acids, but also starch retrogradation, gelatinization of starch in sorghum groats, protein denaturalization in teff flour and melting of lecithin [40,41,42,43,44]. All mixtures showed a statistically insignificant difference in melting curves. However, the packages showed different thermal properties after baking, in each of the three samples 4–6 (Figure 2). The endothermic peak disappeared, which may indicate a lower water content and greater thermal resistance, as the absence of an endothermic file in the 75–82 °C range may mean that there is no non-surface-bound water in the baked packages and may also indicate a lack of retrograde starch [45,46,47]. According to Rolandella’s research, the glass transition temperature [45] decrease along with increase in the water content in the product. This effect is attributed to enthalpy relaxations or structural relaxations found in biopolymer matrix. The baked product may be more stable under certain storage conditions, and should be stored at temperatures below their Tg.

### 3.2. Absorption Properties and Stability of Packaging

An important criterion in determining whether a material can be used as disposable food packaging is its durability, including its ability to resist dropping and low water absorption. For this reason, this research was focused on exploring the packaging durability by drop test. In addition, water absorption capacity was examined. The results of this study showed that the addition of beeswax did not improve nor worsen the structural characteristics of the disposable dishes. Interestingly, the packages made of the OTSL/OTSL + B mixture were not damaged throughout the storage period (Table 4), while OTL and OSL without/with the addition of beeswax were not damaged until 2 weeks after storage. After 2 weeks, disposable dishes dropped on the floor cracked and split. The OTS mixture was the worst and the vessel was destroyed immediately. It can be thus assumed that lecithin improved the plasticity of the flour and increased the durability of the single-dose dishes. The performed stress test correlates with the flexural strength analysis. Buxoo and Jeetah, studied the resistance to dropping of disposable biodegradable paper cups made of pineapple peel, orange peel and Mauritian hemp leaves, coated with a beeswax coating [33]. The use of orange peel and pineapple in the flash test showed that most of the molded cups remained intact, with no cracks. The exception were composites with a higher proportion of orange peel. Buxoo and Jeetah’s research showed that fruit waste is a promising material for disposable tableware [33].

Moreover, the water activity of the prepared packages was checked and the results are presented in Table 4. All packages were characterized by low water activity. Low water activity value suggests that microbial growth is inhibited. The low water activity value and resistance to water absorption indicate that finished and baked packages can be stored under normal conditions in places that are not excessively moist. The packaging showed low water activity, i.e., less than 0.17, and the lowest water activity was found in the samples made of the OTSL/OTSL + B mixture. The low water activity of disposable tableware used as direct food packaging preserves food quality and protects against the development of undesirable microorganisms’ growth [48]. Populations of yeasts such as Candida, Pichia and fungi can be present in fresh olive pomace. However, this is dependent on the variety of the olives as well as the way the olive oil is produced, as it is important to use high-quality raw materials [49]. The pomace is additionally subjected to high temperature, which degrades the bacterial microflora. Additionally, the low activity of water in the final product has a positive effect on the shelf-life of the product.

The disposable food packaging designed in the study the was constructed in the shape of a bowl, a plate and a cup. The cups and bowls have been additionally secured with beeswax to increase their resistance when in contact with water. The cups and bowls were analyzed with and without beeswax protection and tested for their reaction to cold and hot water contact. The results with cold and hot water gave a comparable effect. Therefore, the results are presented in Table 5 in the meantime. 

The water resistance test was carried out until the first drop appeared, leaking through the cup/bowl. It was observed that the addition of beeswax increased the water resistance of the cups from 0.5 h to 1 h. The results of the water resistance test correlate with the decreased water absorption into the package. Disposable utensils prepared from the OTS/OTS + B mix were statistically significantly (*p* < 0.05) less resistant to contact with water and the weight of the packaging increased by half of its initial weight. The OSL packaging was characterized by slightly better parameters, which without the addition of beeswax leaked after about 2.5 h with an addition during about 3 h; however, the packaging after 6 h still increased by 1 g. The mix of OTS and OSL with and without the addition of beeswax is more suitable for dressings than for drinks or soups. It was observed that the best mixes for handling liquid meals or hot drinks were OTL and OTSL (4–4.5 h) with the addition of beeswax. However, without this they also fulfilled their role (3–3.5 h until the first drop appeared). An increase in the weight of the cups/bowls was observed only after 6 h, supporting a possible use for serving hot meals and drinks, especially as 3 h would be sufficient time to consume a hot/cold meal or a coffee/tea drink. Noteworthy is that water resistance tests were carried out with both cold and hot water and gave comparable results, so the packs could be used for storing and reheating meals. The water absorption factor is very important; as a general rule, once the temperature increases, the water is being more quickly [50]. Teff flour is characterized by higher water absorbability and gelling properties, which correlates with the results of DSC before baking packaging [51]. The gelling properties allowed plasticization of the mass and made it easier to form the vessels. Teff also promotes oil absorption, therefore it can be assumed that, when combining the mixture with olive waste, it can absorb fat residues and consequently emulsify more strongly [52]. On the other hand, sorghum groats are characterized by the content of caffeine proteins, which is highly hydrophobic. The heat treatment of sorghum grits reduces its water activity, while increasing water absorption [53,54]. This suggests that the combination of olive pomace and teff flour and/or sorghum, although initially increasing water absorption and viscosity of the product, reduced the water absorption properties of both cold and hot water after baking. The addition of a beeswax coating additionally lowers the value of moisture absorption and gives it hydrophobic properties, which allows extension of the time for keeping the liquid in the cups, even when using hot water [55].

Iewkittayakorn et al. showed that paper cups made of pineapple leaf pulp coated with beeswax showed a longer water-uptake time than uncoated ones. The use of this coating increased the emulsifiability [34]. According to Buxoo and Jeetah, the thickness of the beeswax coating determines its water-repellent properties. A coating area of 0.7 mm resulted in resistance of the cups to leakage for up to 30 min [33]. In our case, the times were longer, which may indicate a more even distribution of the beeswax layer, and the use of olive pomace emulsified better with lipids than the use of pineapple/orange peels.

### 3.3. Nutritional Properties

The developed packaging is potentially edible. Therefore, it was verified whether the antioxidant properties were retained after the production of packaging from the by-products of olive oil production. There are still few studies on consumer biodegradable packaging and its antioxidant properties [33,48,56]. A description of the production of films with a high content of polyphenol compounds can be found in the literature [57,58]. On the other hand, research on disposable tableware is mainly related to mechanical and structural properties [33,48,56]. Still little research can be found on the antioxidant content of disposable tableware [29,37]. For this purpose, the content of polyphenols and the properties of scavenging free radicals were examined. The results are presented in Table 6.

The content of polyphenols ranged from 6.18 to 11.16 mg GAE/100 g, for the 8 weeks stored OSL and the OTSL + B after baking, respectively. The mixtures of OTSL + B (11.16 mg GAE/100 g) and OTSL (10.95 mg GAE/100 g) had the highest content of polyphenols. There is a low decrease in polyphenols with storage time. Noticeably, the addition of teff flour significantly influences the content of polyphenols in the tested products. The addition of polyphenol-rich flour has an enriching effect on the final product, e.g., the addition of grape flour increases the level of polyphenols and antioxidant properties, but the heating process at temperatures above 180 °C reduces polyphenols and antioxidant properties even above 50% [56]. OSL/OSL + B packaging has about half the content of polyphenols but also degrades more quickly during storage. Coating the packages with beeswax had a positive effect on the content of polyphenols in the developed packages and also reduced the degree of polyphenol degradation during storage [59]. The TPC content depends on many factors and is not an exact method, because it also shows other components than just polyphenols, therefore, the TPC content was increased when storing products with a beeswax coating. The increase in TPC could also be influenced by the decomposition of polyphenols contained in beeswax [34,55,59]. The content of phenolic compounds in the olive’s pulp is about 2–3%, while olive oil contains about 50–60 mg/100 g of polyphenols [60,61]. Therefore, a conclusion that the use of olive oil pomace preserved the high phenolic content after heat treatment can be drawn. The content of polyphenols correlates with antioxidant activity (Table 6). OTSL + B had the highest antioxidant potential, the dose of 0.95 mmol TE/100 g reducing free radicals by half. The oxidation properties decreased during storage. Dordevic et al. showed that biodegradable teaspoons from various types of flour, i.e., wheat flour, millet smooth flour and grape flour, lose a significant content of polyphenols, and the antioxidant properties of the designed teaspoons decrease [56].

### 3.4. Biodegradability Properties

All types of the produced packaging samples were subjected to the biodegradation process. The samples were placed in the ground and in the pot and then exposed to the outside to reproduce various weather conditions. Every week, the samples were weighed after prior cleaning and stabilization of the weight in the exicator to drain humidity. After weighing, the test was replaced in the ground and staged outside. Weather conditions were 20–35 °C and moderate rain 6 times per month. The analysis was made for packaging after baking and after 8 weeks of storage. The samples after baking, which were biodegradable within 4 weeks, had reduced their weight by 80%. The products with beeswax coating were the most biodegradable. This may be due to the faster breakdown of beeswax compared to olive pomace. During the further degradation of the samples, those without beeswax degraded by 100%, while those with the beeswax coating by about 90%. The samples with the powder coating degraded completely by week 7. The result of the decrease in masses and the biodegradation process is shown in Figure 3 and Figure 4.

It is worth noting that the storage of the developed disposable tableware does not adversely affect the biodegradation properties. Comparing our product to other biodegradable plant-based products, biodegradation was at a good level. According to research by Iewkittayakorn and others, paper made from pineapple leaf pulp degraded for up to 60 days [34]. In Buxoo and Jeetah biodegradable wax-coated hemp-pineapple composite cup disappeared after 5 weeks in the soil [33]. According to research by Chaabane et al., packaging from sugar cane, palm leaf and paper items is totally degraded, but palm leaf items are degraded by 65% [62].

## 4. Conclusions and Future Perspectives

The study found that disposable packaging such as a plates, cups or bowls can be made from olive oil by-products. The use of olive pomace allows the management and the reduction of landfills. Additively designed disposable tableware can reduce the consumption of plastic products, or products whose manufacturing is not environmentally friendly. The addition of teff flour, sorghum and lecithin enriched the product with valuable polyphenols, increasing the antioxidant potential of the designed products. Disposable dishes can be used for serving cold or hot drinks, soups and for serving dry products. Importantly, the storage of the produced disposable packaging does not affect the characteristics of the product. The developed products exhibit a potential implementation in the following articles and will be explored in further research on the properties of packaging as well as the potential human consumption of the designed packaging. The mixture will also be prepared using the thermos-pressing method, which is less energy-consuming.

## 5. Patents

National patent application 437771.

## Figures and Tables

**Figure 1 foods-11-03776-f001:**
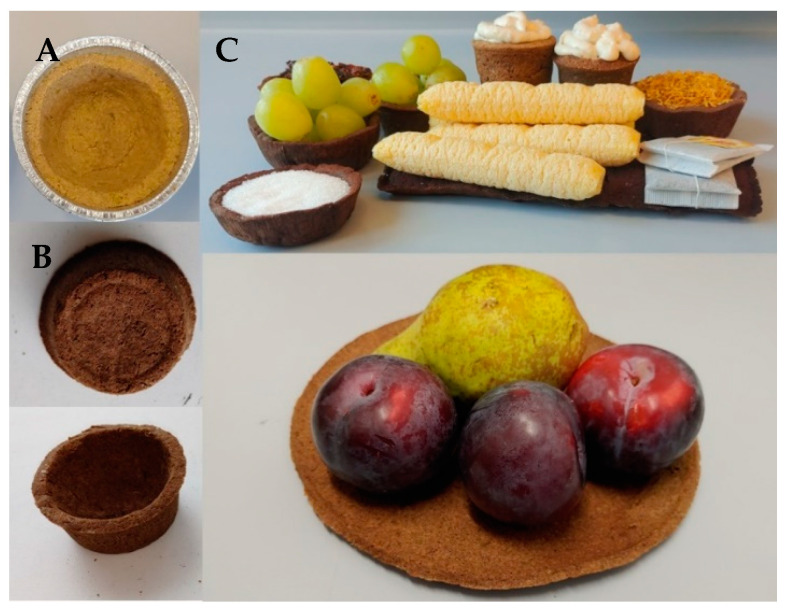
(**A**)—formed disposable utensils before baking; (**B**)—baked disposable utensils; (**C**)—finished disposable dishes in use (own photos).

**Figure 2 foods-11-03776-f002:**
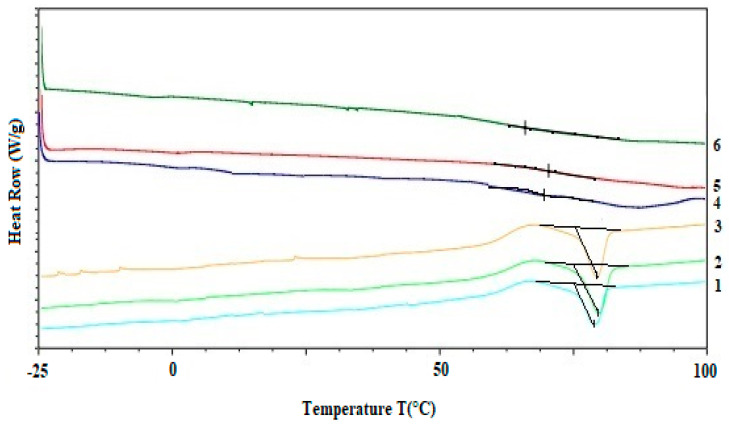
DSC curves of samples before (1—OTSL; 2—OTL; 3—OSL) and after (4—OTSL; 5—OTL; 6—OSL) baking.

**Figure 3 foods-11-03776-f003:**
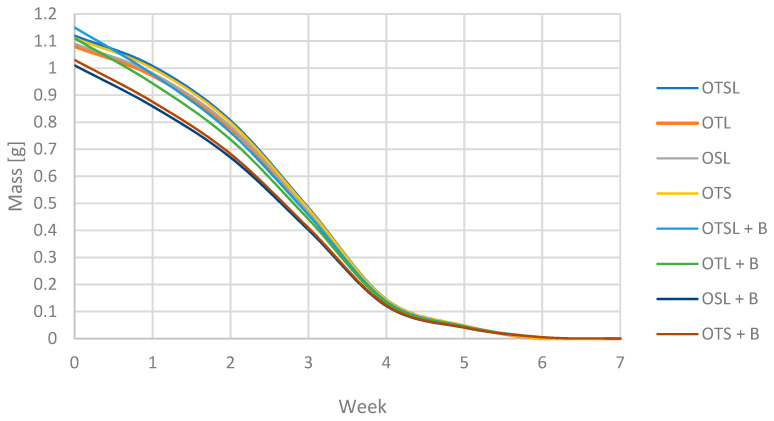
Change in the mass of biodegradable products after baking.

**Figure 4 foods-11-03776-f004:**
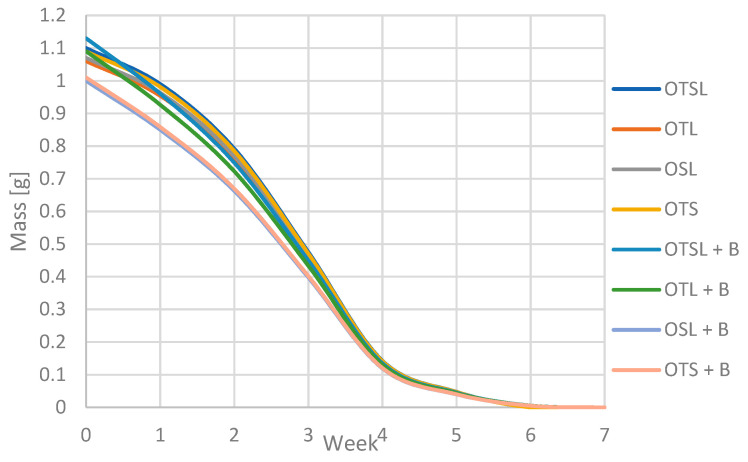
Change in the mass of biodegradable products 8 weeks of storage.

**Table 1 foods-11-03776-t001:** Composition of disposable tableware mixtures.

Mixture	Olive Pomace	Teff Flour	Sorghum Groats	Lecithin
OTSL	+	+	+	+
OTL	+	+	−	+
OSL	+	−	+	+
OTS	+	+	+	−

OTSL: utensils from the mixture of olive pomace, teff flour, sorghum and lecithin; OTL: utensils from the mixture of olive pomace, teff flour and lecithin; OSL: utensils from the mixture of olive pomace, sorghum and lecithin; OTS: utensils from the mixture of olive pomace, teff flour and sorghum. “+”—added ingredient; “−”—no ingredient.

**Table 2 foods-11-03776-t002:** The flexural strength of the disposable utensils.

Mixture	Flexural Strength [N/mm^2^]				
	After Baking	1 Week	2 Weeks	4 Weeks	8 Weeks
OTSL	10.25 ± 0.01 ^a^	10.28 ± 0.02 ^a^	10.29 ± 0.01 ^a^	10.31 ± 0.02 ^a^	11.45 ± 0.02 ^d^
OTL	10.02 ± 0.02 ^a^	10.03 ± 0.01 ^a^	10.31 ± 0.02 ^a^	10.55 ± 0.01 ^a^	12.01 ± 0.03 ^e^
OSL	10.33 ± 0.01 ^a^	10.35 ± 0.02 ^a^	10.39 ± 0.02 ^a^	10.98 ± 0.01 ^a^	12.22 ± 0.02 ^e^
OTS	7.45 ± 0.01 ^b^	8.49 ± 0.02 ^c^	9.01± 0.01 ^c^	10.82 ± 0.02 ^a^	19.21 ± 0.04 ^f^
OTSL + B	10.28 ± 0.02 ^a^	10.31 ± 0.02 ^a^	10.33 ± 0.02 ^a^	10.55 ± 0.02 ^a^	11.59 ± 0.03 ^d^
OTL + B	10.04 ± 0.01 ^a^	10.12 ± 0.02 ^a^	10.48 ± 0.02 ^a^	10.64 ± 0.02 ^a^	12.25 ± 0.01 ^e^
OSL + B	10.36 ± 0.02 ^a^	10.39 ± 0.01 ^a^	10.95 ± 0.01 ^a^	11.01 ± 0.01 ^d^	12.18 ± 0.02 ^e^
OTS + B	7.49 ± 0.01 ^b^	8.55 ± 0.01 ^c^	9.88 ± 0.03 ^c^	11.05 ± 0.01 ^d^	19.59 ± 0.03 ^f^

^a–f^ Same superscript letter in one row indicates no statistically significant differences between mixture (*p* < 0.05). OTSL + B: OTSL covered with beeswax; OTL + B: OTL covered with beeswax; OSL + B: OSL covered with beeswax; OTS + B: OTS covered with beeswax.

**Table 3 foods-11-03776-t003:** Parameters for measuring the color of disposable packaging.

Mixture	Without the Addition of Beeswax			With the Addition of Beeswax		
	a*	b*	L*	a*	b*	L*
**After baking**						
OTSL	3.31 ± 0.01 ^a^	7.56 ± 0.02 ^c^	38.51 ± 0.06 ^e^	3.30 ± 0.01 ^a^	7.57 ± 0.01 ^c^	38.59 ± 0.05 ^e^
OTL	3.65 ± 0.02 ^a^	10.00 ± 0.02 ^d^	41.56 ± 0.04 ^f^	3.62 ± 0.01 ^a^	10.02 ± 0.02 ^d^	41.62 ± 0.03 ^f^
OSL	3.77 ± 0.01 ^a^	10.28 ± 0.01 ^d^	42.44 ± 0.05 ^f^	3.74 ± 0.01 ^a^	10.29 ± 0.02 ^d^	42.49 ± 0.03 ^f^
OTS	4.03± 0.01 ^b^	9.05 ± 0.01 ^c,d^	39.40 ± 0.06 ^e^	4.00± 0.01 ^b^	9.06 ± 0.01 ^c,d^	39.45 ± 0.05 ^e^
**1 week**						
OTSL	3.32 ± 0.01 ^a^	7.55 ± 0.02 ^c^	37.50 ± 0.03 ^e^	3.33 ± 0.01 ^a^	7.56 ± 0.02 ^c^	37.68 ± 0.03 ^e^
OTL	3.72 ± 0.01 ^a^	10.45 ± 0.01 ^d^	42.31 ± 0.03 ^f^	3.73 ± 0.01 ^a^	10.47 ± 0.02 ^d^	42.87 ± 0.04 ^f^
OSL	3.81 ± 0.01 ^a^	10.32 ± 0.01 ^d^	43.18 ± 0.04 ^f^	3.84 ± 0.02 ^a^	10.33 ± 0.03 ^d^	43.65 ± 0.03 ^f^
OTS	4.03 ± 0.01 ^b^	9.18 ± 0.02 ^c,d^	39.17 ± 0.04 ^e^	4.05 ± 0.01 ^b^	9.20 ± 0.02 ^c,d^	39.28 ± 0.06 ^e^
**2 weeks**						
OTSL	3.32 ± 0.01 ^a^	7.56 ± 0.01 ^c^	37.55 ± 0.02 ^e^	3.34 ± 0.01 ^a^	7.59 ± 0.01 ^c^	37.71 ± 0.04 ^e^
OTL	3.72 ± 0.01 ^a^	10.45 ± 0.01 ^d^	42.31 ± 0.03 ^f^	3.74 ± 0.02 ^a^	10.48 ± 0.02 ^d^	42.92 ± 0.03 ^f^
OSL	3.81 ± 0.01 ^a^	10.32 ± 0.01 ^d^	43.18 ± 0.04 ^f^	3.86 ± 0.01 ^a^	10.34 ± 0.02 ^d^	43.99 ± 0.04 ^f^
OTS	4.06 ± 0.02 ^b^	7.40 ± 0.01 ^c^	37.10 ± 0.03 ^e^	4.08 ± 0.02 ^b^	7.42 ± 0.02 ^c^	37.45 ± 0.02 ^e^
**3 weeks**						
OTSL	3.33 ± 0.01 ^a^	7.58 ± 0.01 ^c^	37.59 ± 0.02 ^e^	3.34 ± 0.02 ^a^	7.62 ± 0.02 ^c^	37.72 ± 0.04 ^e^
OTL	3.83 ± 0.02 ^a^	10.62 ± 0.03 ^d^	42.67 ± 0.03 ^f^	3.85 ± 0.02 ^a^	10.68 ± 0.03 ^d^	43.01 ± 0.02 ^f^
OSL	3.95 ± 0.02 ^a^	10.77 ± 0.02 ^d^	42.97 ± 0.02 ^f^	3.99 ± 0.02 ^a^	10.81 ± 0.03 ^d^	44.11 ± 0.06 ^f^
OTS	4.11 ± 0.03 ^b^	10.98 ± 0.05 ^d^	40.22 ± 0.05 ^f^	4.15 ± 0.01 ^b^	11.01 ± 0.02 ^d^	44.01 ± 0.03 ^f^
**4 weeks**						
OTSL	3.36 ± 0.01 ^a^	7.71 ± 0.02 ^c^	37.84 ± 0.01 ^e^	3.38 ± 0.02 ^a^	7.71 ± 0.03 ^c^	37.89 ± 0.04 ^e^
OTL	3.92 ± 0.01 ^a^	10.78 ± 0.01 ^d^	43.75 ± 0.02 ^f^	3.99 ± 0.02 ^a^	10.78 ± 0.02 ^d^	43.22 ± 0.02 ^f^
OSL	4.01 ± 0.01 ^b^	11.02 ± 0.03 ^d^	43.12 ± 0.03 ^f^	4.18 ± 0.02 ^b^	11.02 ± 0.03 ^d^	44.18 ± 0.05 ^f^
OTS	4.18 ± 0.02 ^b^	11.18 ± 0.02 ^d^	42.28 ± 0.03 ^f^	4.21 ± 0.02 ^b^	11.18 ± 0.02 ^d^	44.22 ± 0.02 ^f^
**8 weeks**						
OTSL	3.39 ± 0.02 ^a^	7.70 ± 0.01 ^c^	37.87 ± 0.02 ^e^	3.41 ± 0.03 ^a^	7.75 ± 0.02 ^c^	37.93 ± 0.03 ^e^
OTL	4.02 ± 0.03 ^b^	11.03 ± 0.03 ^d^	42.95 ± 0.03 ^f^	4.06 ± 0.02 ^b^	11.09 ± 0.03 ^d^	43.25 ± 0.05 ^f^
OSL	4.12 ± 0.02 ^b^	11.45 ± 0.02 ^d^	43.98 ± 0.04 ^f^	4.19 ± 0.01 ^b^	11.52 ± 0.01 ^d^	44.25± 0.04 ^f^
OTS	4.24 ± 0.02 ^b^	8.90 ± 0.01 ^c,d^	39.64 ± 0.04 ^e^	4.31 ± 0.02 ^b^	8.91 ± 0.02 ^c,d^	43.61± 0.06 ^e^

^a–f^ Same superscript letter in one row indicates no statistically significant differences between mixtures (*p* < 0.05).

**Table 4 foods-11-03776-t004:** The results of crack analysis of disposable utensils after drop test and water activity.

Mixture	Crack Analysis				
	After Baking	1 Week	2 Weeks	4 Weeks	8 Weeks
OTSL	remained intact	remained intact	remained intact	remained intact	remained intact
OTL	remained intact	remained intact	remained intact	cracked/split	cracked/split
OSL	remained intact	remained intact	remained intact	cracked/split	cracked/split
OTS	cracked/split	cracked/split	cracked/split	cracked/split	cracked/split
OTSL + B	remained intact	remained intact	remained intact	remained intact	remained intact
OTL + B	remained intact	remained intact	remained intact	cracked/split	cracked/split
OSL + B	remained intact	remained intact	remained intact	cracked/split	cracked/split
OTS + B	cracked/split	cracked/split	cracked/split	cracked/split	cracked/split
	**Water Activity**				
OTSL	0.059 ± 0.01 ^a^	0.059 ± 0.01 ^a^	0.060 ± 0.01 ^a^	0.059 ± 0.02 ^a^	0.052 ± 0.2 ^a^
OTL	0.069 ± 0.02 ^a^	0.070 ± 0.01 ^b^	0.072 ± 0.02 ^b^	0.082 ± 0.02 ^b^	0.089 ± 0.02 ^b^
OSL	0.055 ± 0.01 ^a^	0.056 ± 0.01 ^a^	0.057 ± 0.02 ^a^	0.056 ± 0.01 ^a^	0.052 ± 0.02 ^a^
OTS	0.098 ± 0.01 ^b^	0.170 ± 0.01 ^c^	0.175 ± 0.02 ^c^	0.198 ± 0.03 ^c^	0.115 ± 0.05 ^c^
OTSL + B	0.057 ± 0.02 ^a^	0.057 ± 0.01 ^a^	0.060 ± 0.02 ^a^	0.061 ± 0.01 ^a^	0.062 ± 0.02 ^a^
OTL + B	0.068 ± 0.02 ^a^	0.068 ± 0.02 ^b^	0.069 ± 0.02 ^b^	0.077 ± 0.01 ^b^	0.081 ± 0.01 ^b^
OSL + B	0.054 ± 0.01 ^a^	0.061 ± 0.01 ^a^	0.061 ± 0.02 ^a^	0.065 ± 0.01 ^a^	0.066 ± 0.01 ^a^
OTS + B	0.097 ± 0.02 ^b^	0.125 ± 0.03 ^c^	0.159 ± 0.01 ^c^	0.160± 0.03 ^c^	0.162 ± 0.03 ^c^

^a–c^ Same superscript letter in one row indicates no statistically significant differences between mixtures (*p* < 0.05).

**Table 5 foods-11-03776-t005:** Occurrence of first drop and water absorption by the package.

Mixture		The First Drop (h)		
		After Baking	4 Weeks	8 Weeks
OTSL		3.5 ± 0.01 ^a^	3.5 ± 0.01 ^a^	3.5 ± 0.01 ^a^
OTL		3 ± 0.01 ^a^	3.2 ± 0.01 ^a^	3.2 ± 0.01 ^a^
OSL		2.5 ± 0.01 ^b^	2.5 ± 0.01 ^b^	2.4 ± 0.01 ^b^
OTS		0.5 ± 0.01 ^c^	0.5 ± 0.01 ^c^	0.2 ± 0.01 ^c^
OTSL + B		4.5 ± 0.01 ^d^	4.5 ± 0.01 ^d^	4.5 ± 0.01 ^d^
OTL + B		4.1 ± 0.01 ^d^	4.1 ± 0.01 ^d^	4.1 ± 0.01 ^d^
OSL + B		3 ± 0.01 ^a^	3 ± 0.01 ^a^	3 ± 0.01 ^a^
OTS + B		1 ± 0.01 ^c^	1 ± 0.01 ^c^	1 ± 0.01 ^c^
	**Weight before the test [g]**	**Weight after 6 h [g]**		
OTSL	2.21 ± 0.02 ^a^	2.26 ± 0.01 ^a^	2.32 ± 0.02 ^b^	2.36 ± 0.01 ^b^
OTL	2.22 ± 0.01 ^a^	2.51 ± 0.04 ^b^	2.53 ± 0.05 ^b^	2.55 ± 0.03 ^b^
OSL	2.36 ± 0.01 ^a^	3.62 ± 0.03 ^c^	3.59 ± 0.04 ^c^	3.64 ± 0.02 ^c^
OTS	2.44 ± 0.01 ^a^	6.49 ± 0.04 ^d^	6.48 ± 0.03 ^d^	6.51 ± 0.04 ^d^
OTSL + B	2.22 ± 0.02 ^a^	2.24 ± 0.01 ^a^	2.24 ± 0.02 ^a^	2.24 ± 0.01 ^a^
OTL + B	2.23 ± 0.02 ^b^	2.44 ± 0.01 ^b^	2.46 ± 0.02 ^b^	2.48 ± 0.02 ^b^
OSL + B	2.37 ± 0.01 ^a^	3.01 ± 0.02 ^c^	3.05 ± 0.03 ^c^	3.02 ± 0.01 ^c^
OTS + B	2.45 ± 0.02 ^a^	4.66 ± 0.03 ^d^	4.29 ± 0.01 ^d^	4.78 ± 0.02 ^d^

^a–d^ Same superscript letter in one row indicates no statistically significant differences between mixtures (*p* < 0.05).

**Table 6 foods-11-03776-t006:** Total polyphenol content and antioxidant activity.

Mixture	DPPH (mmol TE/100 g)				
	After Baking	1 Week	2 Weeks	4 Weeks	8 Weeks
OTSL	1.11 ± 0.01 ^a^	1.25 ± 0.02 ^a^	1.66 ± 0.03 ^a^	1.89 ± 0.04 ^a^	2.92 ± 0.01 ^a^
OTL	3.09 ± 0.02 ^b^	3.19 ± 0.01 ^b^	3.72 ± 0.02 ^b^	3.88 ± 0.03 ^b^	5.09 ± 0.03 ^c^
OSL	4.11 ± 0.01 ^a^	4.36 ± 0.05 ^a^	4.57 ± 0.02 ^a^	4.85 ± 0.03 ^a^	7.50 ± 0.03 ^b^
OTS	3.15 ± 0.03 ^b^	3.23 ± 0.01 ^b^	3.75 ± 0.02 ^b^	3.91 ± 0.02 ^b^	6.98 ± 0.03 ^c^
OTSL + B	0.95 ± 0.01 ^a^	0.98 ± 0.03 ^a^	1.21 ± 0.02 ^a^	1.33 ± 0.02 ^a^	1.85 ± 0.03 ^a^
OTL + B	2.89 ± 0.02 ^b^	2.95 ± 0.01 ^b^	3.16 ± 0.03 ^b^	3.44 ± 0.04 ^b^	4.36 ± 0.05 ^c^
OSL + B	3.00 ± 0.06 ^a^	3.15 ± 0.03 ^a^	3.22 ± 0.04 ^a^	3.49 ± 0.03 ^a^	4.37 ± 0.03 ^b^
OTS + B	3.03 ± 0.02 ^b^	3.11 ± 0.03 ^b^	3.37 ± 0.03 ^b^	3.69 ± 0.04 ^b^	5.87 ± 0.04 ^c^
	**Total polyphenol content [mg GAE/100 g]**				
OTSL	10.95 ± 0.01 ^a^	10.91 ± 0.02 ^a^	10.89 ± 0.01 ^a^	10.86 ± 0.01 ^a^	10.82 ± 0.02 ^a^
OTL	10.44 ± 0.01 ^b^	10.38 ± 0.01 ^b^	10.35 ± 0.01 ^b^	10.30 ± 0.01 ^b^	10.25 ± 0.01 ^b^
OSL	8.36 ± 0.01 ^b^	8.31 ± 0.02 ^b^	6.28 ± 0.01 ^b^	6.24 ± 0.01 ^b^	6.18 ± 0.01 ^c^
OTS	10.95 ± 0.01 ^a^	10.90 ± 0.01 ^a^	10.84 ± 0.01 ^a^	10.66 ± 0.01 ^a^	10.39 ± 0.01 ^a^
OTSL + B	11.16 ± 0.03 ^a^	11.12 ± 0.02 ^a^	11.04 ± 0.02 ^a^	10.59 ± 0.02 ^a^	10.54 ± 0.01 ^a^
OTL + B	10.66 ± 0.02 ^b^	10.59 ± 0.01 ^b^	10.52 ± 0.01 ^b^	10.49 ± 0.01 ^b^	10.47 ± 0.01 ^b^
OSL + B	9.55 ± 0.03 ^b^	9.52 ± 0.01 ^b^	9.48 ± 0.01 ^b^	9.31 ± 0.01 ^b^	9.25 ± 0.01 ^c^
OTS + B	10.03 ± 0.02 ^a^	10.58 ± 0.03 ^a^	10.57 ± 0.03 ^a^	10.56 ± 0.01 ^a^	10.49 ± 0.02 ^a^

^a–c^ Same superscript letter in one row indicates no statistically significant differences between mixtures (*p* < 0.05).

## Data Availability

Data is contained within the article.

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
