# Peer review of "Quality Assessment of Waste from Olive Oil Production and Design of Biodegradable Packaging"

_foods, 2022, doi:10.3390/foods11233776_

Round 1

Reviewer 1 Report

Dear authors,

The article deals with two key environmental problems for sustainable production: plastic pollution due to single-use tableware and the reduction and valorization of by-products of the food industry. In this work the authors designed and studied fully biobased materials proposed as edible and disposable tableware. It should be noted the novel approach of this material in using an agri-food industry by-product, such as olive oil pomace, as raw material highlighting its antioxidant activity. Even though the article falls into the aims and scopes of the journal, there are some experimental data missing among with further results discussions. Please take into account the following listed comments when revising your work:

1.     In the Introduction section it reads: “Currently, disposable tableware is produced from various types of biodegradable polymers (starch, polyhydroxyalkanoate, polyethylene with starch) obtained from various plant products or microorganisms [18-21]. However, such production also generates high carbon dioxide emissions to the atmosphere [22,23]. Therefore, new packaging materials are being extensively explored, and other agricultural products such as bran and corn are recently in use.” Literature review of bioplastics Life Cylce Assesments (LCAs) indicate that in general biobased plastic materials such as PLA, starch-based and other bioplstics have significantly lower greenhouse gases GHG emissions than the fossil oil counterparts, mostly attributed to plants CO2 absorption in photosynthesis before harvest. Therefore, it is not clear how other agricultural products for bioplastic production would change such effect. Please clarify

Come references recommended regarding bioplastics LCAs:

F. Razza, F.D. Innocenti, A. Dobon et al., Environmental profle

of a bio-based and biodegradable foamed packaging prototype

in comparison with the current benchmark. Journal of Cleaner Production 102, 493–500 (2015). https://doi.org/10.1016/j.jclepro.2015.04.033

S. Spierling, E. Knüpfer, H. Behnsen et al., Bio-based plastics -a review of environmental, social and economic impact assessments. Journal of Cleaner Production 185, 476–491 (2018). https://doi.org/10.1016/j.jclepro.2018.03.014

R. Shogren, D. Wood, W. Orts, G. Glenn, Plant-based materials and transitioning to a circular economy. Sustainable Production and Consumption 19, 194–215 (2019). https://doi.org/10.1016/j.spc.2019.04.007

G. Espadas-Aldana, P. Guaygua-Amaguaña, C. Vialle, JP. Belaud, P. Evon, C. Sablayrolles, Life Cycle Assessment of Olive Pomace as a Reinforcement in Polypropylene and Polyethylene Biocomposite Materials: A New Perspective for the Valorization of This Agricultural By-Product. Coatings 11(5), 525 (2021). https://doi.org/10.3390/coatings11050525

2.     Raw materials origin and especially olive pomace origin and characterization are required. Perhaps this work is useful for results comparison: S. Lammi, N. Le Moigne, D. Djenane, N. Gontard, H.Angellier-Coussy, Dry fractionation of olive pomace for the development of food packaging biocomposites. Industrial Crops and Products 120, 250-261 (2018). https://doi.org/10.1016/j.indcrop.2018.04.052

3.     Tray mold characteristics (dimensions, material, etc.) and weight/volume ratio of the formulated mass should be included, was shrinkage or expansion observed after drying process? Is the density of the material adequate for its purpose? Considering that low-weight materials are preferable for disposable tableware to minimize transportation cost and easy handling.

4.     The baking process appears to be high energy consuming. Have the authors considered other processing technologies, such as thermos-pressing? It would be interesting to include a discussion on the sustainability of such requirements for future research on these materials.

5.     It is important to indicate storage conditions under which materials were preserved for shelf-life evaluation.

6.     The cited work of Jüri et al [24] (line 89) in the Flexural Strength section is not listed in the References section. Please include it and/or specify within the text the modifications in the method.

7.     How many replicates were tested for flexural and crack resistance test and biodegradation under burial analysis?

8.     In the drop test were cracks detected by visual inspection?

9.     Under what relative humidity conditions were samples equilibrated for water activity assessment?

10.  In Table 2 the superscript letters used for ANOVA significant differences is not clear, typically ascendent or descendent values are used, is the comparison among treatments or storage time or the combination of both? Please clarify.

11.  The flexural test results show an increment in flexural strength with storage time, especially in the las period. This could be, for instance, due to starch retrogradation in the composite material. Beside OTS material showed the lowest mechanical resistance within the first month and presented the higher increase in flexural strength after 8 weeks, this may be probably attributed to microstructural differences and interaction within composite compounds, further discussion on these results is recommended. If possible SEM evaluation of composites surface and cross-section and/or FTIR of the materials would help understand such effects.

12.  Please, check the spelling and grammar throughout the whole manuscript. There are some spelling and grammatical mistakes, missing spaces, punctuation marks and words that make reading difficult, check for example, line 191, 215, 216, 229-230, 297 and 378.

13.  The DSC curves before baking show and endothermic peak where it is indicated by the authors as a melting peak: “The DSC curve of the unbaked samples is characterized by a single endothermic peak observed in the temperature range of 75-82 °C. The melting temperature peaks (Tm) of the unbaked OTSL, OTL and OSL samples were observed at temperatures of: 77 °C, (heat capacity, 105 J/g), 79 °C (112 J/g) and 78 °C (109 J/g), respectively.” Why was OTS not measured? Considering the materials composition, it is probable that the endothermic peak is attributed to gelatinization of starch in sorghum groats, protein denaturalization in teff flour and melting of lecithin. See references below:

D. Wang, S. Bean, J. McLaren, P. Seib, R. Madl, M. Tuinstra, Y. Shi, M. Lenz, X. Wu, R. Zhao. Grain sorghum is a viable feedstock for ethanol production. Journal of Industrial Microbiology and Biotechnology 35, 313–320 (2008). https://doi.org/10.1007/s10295-008-0313-1

AR. A. Adebowale, M. Naushad Emmambux, M. Beukes, J.R.N. Taylor, Fractionation and characterization of teff proteins. Journal of Cereal Science 54, 380-386 (2011). https://doi.org/10.1016/j.jcs.2011.08.002

K. Mészáros Szécsény, I. Esztelecki, G. Pokol, Adventages and limits on usage of thermal methods in complex systems: Bread and bread additives analyses. Journal of Thermal Analysis and Calorimetry 89 (3), 829–833 (2007). https://doi.org/10.1007/s10973-006-7928-0  

14.  Later on in the text reads: “The endothermic peak disappeared, which may indicate a lower water content and greater thermal resistance, as the absence of an endothermic file in the 75-82 °C range may mean that there is no non-surface-bound water in the baked packages, and may also indicate a lack of retrograde starch.” Could this transition be attributed to glass transition temperature of the present biopolymers?

15.  In Table 5 some “,” should be changed for “.”

16.  In lines 307-309 the authors claim a better wax distribution over the studied material. Was the surface coverage assessed by any means?

17.  In lines 339-341 please clarify what is the loss of polyphenols and antioxidant capacity in tablespoons with grape flour studied by Dordevic et al. 2021 attributed to.

18.  Regarding the biodegradability assay, with no intention of disregarding the obtained results, which are relevant considering the end of life of the studied material and product. In accordance with the test conditions, it can be stated that the materials are degradable under burial, yet following current standards, as established by the European standards EN 13432 / EN 14995 testing for certification include:

-       Chemical test: Disclosure of all constituents, threshold values for heavy metals are to be adhered to.

-       Biodegradability in controlled composting conditions (oxygen consumption and production of CO2): Proof must be made that at least 90 percent of the organic material is converted into CO2 within 6 months.

-       Disintegration: After 3 months composting and subsequent sifting through a 2 mm sieve, no more than 10 percent residue may remain, as compared to the original mass.

-       Ecotoxicity test: Examination of the effect of resultant compost on plant growth (agronomic test).

19.   Lastly, considering the reported phytotoxicity of olive pomace residues due to its high polyphenols content (Hachicha et al. 2009, Caporaso et al. 2017, Nunes et al. 2019). However as reported by Hachicha et al. 2009, the ecotoxicity of this by-product can be eliminated by co-composting with sesame bark. Given the polyphenols content of the materials reported in this work and the composition of the composite materials. May the carbon-rich compounds of the composite materials help reduce the ecotoxicity of the material during composting process? Do the authors believe an ecotoxicity test would be required?

Salma Hachicha, Juan Cegarra, Fatma Sellami, Ridha Hachicha, Noureddine Drira, K. Medhioub, Emna Ammar, Elimination of polyphenols toxicity from olive mill wastewater sludge by its co-composting with sesame bark. Journal of Hazardous Materials 161, 1131–1139 (2009). https://doi.org/10.1016/j.jhazmat.2008.04.066

N. Caporaso, D. Formisano, A. Genovese, Use of phenolic compounds from. olive mill wastewater as valuable ingredients for functional foods. Critical Reviews in Food Science and Nutrition 58 (16), 2829-2841 (2018). http://dx.doi.org/10.1080/10408398.2017.1343797

M.A. Nunes, S. Pawlowski, A.S.G. Costa, R.C. Alves, M.B.P.P. Oliveira, S. Velizarov, Valorization of olive pomace by a green integrated approach applying sustainable extraction and membrane-assisted concentration.

Science of the Total Environment 652, 40–47 (2019). https://doi.org/10.1016/j.scitotenv.2018.10.204

Hope you find this revision helpful to improve this original work of yours.

Best regards

Author Response

Dear Reviewer,

Thank you for your letter and constructive comments concerning our manuscript entitled “Quality assessment of waste from olive oil production and de-sign of biodegradable packaging”. We have read your comments carefully and made correction according to your suggestions to our manuscript. The most important changes are marked in green.

Thank you very much for your effort.

In the following, we give a point-by-point reply to your comments:

Reviewer 1

  1. In the Introduction section it reads: “Currently, disposable tableware is produced from various types of biodegradable polymers (starch, polyhydroxyalkanoate, polyethylene with starch) obtained from various plant products or microorganisms [18-21]. However, such production also generates high carbon dioxide emissions to the atmosphere [22,23]. Therefore, new packaging materials are being extensively explored, and other agricultural products such as bran and corn are recently in use.” Literature review of bioplastics Life Cylce Assesments (LCAs) indicate that in general biobased plastic materials such as PLA, starch-based and other bioplstics have significantly lower greenhouse gases GHG emissions than the fossil oil counterparts, mostly attributed to plants CO2 absorption in photosynthesis before harvest. Therefore, it is not clear how other agricultural products for bioplastic production would change such effect. Please clarify

Come references recommended regarding bioplastics LCAs:

  1. Razza, F.D. Innocenti, A. Dobon et al., Environmental profle of a bio-based and biodegradable foamed packaging prototype in comparison with the current benchmark. Journal of Cleaner Production 102, 493–500 (2015). https://doi.org/10.1016/j.jclepro.2015.04.033
  2. Spierling, E. Knüpfer, H. Behnsen et al., Bio-based plastics -a review of environmental, social and economic impact assessments. Journal of Cleaner Production 185, 476–491 (2018). https://doi.org/10.1016/j.jclepro.2018.03.014
  3. Shogren, D. Wood, W. Orts, G. Glenn, Plant-based materials and transitioning to a circular economy. Sustainable Production and Consumption 19, 194–215 (2019). https://doi.org/10.1016/j.spc.2019.04.007
  4. Espadas-Aldana, P. Guaygua-Amaguaña, C. Vialle, JP. Belaud, P. Evon, C. Sablayrolles, Life Cycle Assessment of Olive Pomace as a Reinforcement in Polypropylene and Polyethylene Biocomposite Materials: A New Perspective for the Valorization of This Agricultural By-Product. Coatings 11(5), 525 (2021). https://doi.org/10.3390/coatings11050525

Response:

We agree with this comment and the revised version of the manuscript was complemented by this information. Thank you very much for providing a list of publications on this topic, which undoubtedly made it easier for us to make the recommended improvements. In the revised version of the manuscript, we have included the following sentences:

“Starch and bioplastics in the Life Cylce Assessments (LCAs) process, have a reduced generation of carbon dioxide emissions into the atmosphere [20-25]. The use of fresh agri-cultural produce results in higher emissions of carbon dioxide into the atmosphere [26,27]. To reduce the content of carbon dioxide emissions, packaging production time can be shortened, but also plant waste can be used, the deposition of which in landfills in-creases the emission of carbon dioxide [28].”

  1. Raw materials origin and especially olive pomace origin and characterization are required. Perhaps this work is useful for results comparison: S. Lammi, N. Le Moigne, D. Djenane, N. Gontard, H.Angellier-Coussy, Dry fractionation of olive pomace for the development of food packaging biocomposites. Industrial Crops and Products 120, 250-261 (2018). https://doi.org/10.1016/j.indcrop.2018.04.052

Response:

Information on the origin of the olive marc has been added.:

 “The olive pomace is made from green olives of the Manzanilla variety from Spain, cold-pressed. Olive pomace is the residue of pulp and skin, the initial moisture content of which was 51% by weight.”

  1. Tray mold characteristics (dimensions, material, etc.) and weight/volume ratio of the formulated mass should be included, was shrinkage or expansion observed after drying process? Is the density of the material adequate for its purpose? Considering that low-weight materials are preferable for disposable tableware to minimize transportation cost and easy handling.

Response:

We agree with this comment and the revised version of the manuscript was complemented by this information. Samples of the same size must be selected for the strength analysis, therefore the dimension was kept. During drying, the weight of the packages decreased compared to prior to drying, however, the final product is heavier than the plastic packages. We understand that low weight products are preferred for reasons of minimizing transport costs. However, we believe that it is a competitor product due to the potential for the use of precipitation. Market research allows us to state that companies would like to purchase such products, even if it would involve more expensive transport.

This article is a preliminary research to select the appropriate material, work is underway to improve the product.

  1. The baking process appears to be high energy consuming. Have the authors considered other processing technologies, such as thermos-pressing? It would be interesting to include a discussion on the sustainability of such requirements for future research on these materials.

Response:

Thank you for your valuable comment. We agree with the Reviewer that the baking process may appear to be high energy consuming. However, for the time being we have been doing preliminary tests, the material was made in laboratory conditions. We want to use machines found in industry to produce this type of research. Since we did not perform these analyzes, we did not add the results, because the determination of the parameters would only be a suggestion of what could constitute scientific unreliability. Thank you for your suggestion and we will include information in the prospects for further research.

  1. It is important to indicate storage conditions under which materials were preserved for shelf-life evaluation.

Response:

Storage information has been added in section: 2.2. Material and Preparation.

“The all packages were stored at room temperature in a dark place (in a cabinet) without additional packages.”

  1. 6. The cited work of Jüri et al [24] (line 89) in the Flexural Strength section is not listed in the References section. Please include it and/or specify within the text the modifications in the method.

Response:

We thank the Reviewer for the suggestions. This mistake has been corrected.

  1. How many replicates were tested for flexural and crack resistance test and biodegradation under burial analysis?

Response:

 Five replicates were made for the tests: flexural and crack and three repetitions for biodegradability. This information was also included in the methodology of the revised manuscript.

  1. In the drop test were cracks detected by visual inspection?

Response:

Yes in the drop test, cracks were detected by visual inspection.

  1. Under what relative humidity conditions were samples equilibrated for water activity assessment?

Response:

 The samples were closed in special vessels to balance the wetness, and then transferred in these vessels to the device. Vessel moisture ranging from 0.5-2% has been tested in other studies.

  1. In Table 2 the superscript letters used for ANOVA significant differences is not clear, typically ascendent or descendent values are used, is the comparison among treatments or storage time or the combination of both? Please clarify.

Response:

A statistically significant differences were determined by combining both of these features to make the differences more apparent.

  1. The flexural test results show an increment in flexural strength with storage time, especially in the las period. This could be, for instance, due to starch retrogradation in the composite material. Beside OTS material showed the lowest mechanical resistance within the first month and presented the higher increase in flexural strength after 8 weeks, this may be probably attributed to microstructural differences and interaction within composite compounds, further discussion on these results is recommended. If possible SEM evaluation of composites surface and cross-section and/or FTIR of the materials would help understand such effects.

 Response:

We would like to thank the Reviewer for this helpful suggestion. In the next stage of the research, we want to perform the SEM and FTIR analysis, but we also want to define the TG analysis together with the connection with MS to be able to accurately characterize the composition of our packaging. At this stage, it is difficult for us to prove whether it is a change only due to the retrogradation of starch or it may be related to the high content of fat in olive pomace. According to DSC, the retrograde of starch has decreased, so we need to look closely at thermal transformation. We are also considering combining DSC with TG and using XRD.

  1. Please, check the spelling and grammar throughout the whole manuscript. There are some spelling and grammatical mistakes, missing spaces, punctuation marks and words that make reading difficult, check for example, line 191, 215, 216, 229-230, 297 and 378.

 Response:

The manuscript has been checked and corrected by a native speaker.

  1. The DSC curves before baking show and endothermic peak where it is indicated by the authors as a melting peak: “The DSC curve of the unbaked samples is characterized by a single endothermic peak observed in the temperature range of 75-82 °C. The melting temperature peaks (Tm) of the unbaked OTSL, OTL and OSL samples were observed at temperatures of: 77 °C, (heat capacity, 105 J/g), 79 °C (112 J/g) and 78 °C (109 J/g), respectively.” Why was OTS not measured? Considering the materials composition, it is probable that the endothermic peak is attributed to gelatinization of starch in sorghum groats, protein denaturalization in teff flour and melting of lecithin. See references below:

  1. Wang, S. Bean, J. McLaren, P. Seib, R. Madl, M. Tuinstra, Y. Shi, M. Lenz, X. Wu, R. Zhao. Grain sorghum is a viable feedstock for ethanol production. Journal of Industrial Microbiology and Biotechnology 35, 313–320 (2008). https://doi.org/10.1007/s10295-008-0313-1
  2. A. Adebowale, M. Naushad Emmambux, M. Beukes, J.R.N. Taylor, Fractionation and characterization of teff proteins. Journal of Cereal Science 54, 380-386 (2011). https://doi.org/10.1016/j.jcs.2011.08.002
  3. Mészáros Szécsény, I. Esztelecki, G. Pokol, Adventages and limits on usage of thermal methods in complex systems: Bread and bread additives analyses. Journal of Thermal Analysis and Calorimetry 89 (3), 829–833 (2007). https://doi.org/10.1007/s10973-006-7928-0

Response:

OTS came out the worst in most of the studies, so it was rejected at this stage of the study. This mixture was the least favorable. Thank you very much, we have added a description and references.

  1. Later on in the text reads: “The endothermic peak disappeared, which may indicate a lower water content and greater thermal resistance, as the absence of an endothermic file in the 75-82 °C range may mean that there is no non-surface-bound water in the baked packages, and may also indicate a lack of retrograde starch.” Could this transition be attributed to glass transition temperature of the present biopolymers?

Response:

It could also mean that. We have added a description. “The endothermic peak disappeared, which may indicate a lower water content and greater thermal resistance, as the absence of an endothermic file in the 75-82 °C range may mean that there is no non-surface-bound water in the baked packages and may also indicate a lack of retrograde starch [45-47]. According to Rolandella's research, the glass tran-sition temperature [45] increases with the increase in the water content in the product. Low water content and lower Tg indicate that the baked product may be more stable un-der storage conditions."

  1. 15. In Table 5 some “,” should be changed for “.”

Response:

Thank you very much for this suggestion. It was corrected.

  1. In lines 307-309 the authors claim a better wax distribution over the studied material. Was the surface coverage assessed by any means?

Response:

The distribution of the wax was evaluated only visually.

  1. In lines 339-341 please clarify what is the loss of polyphenols and antioxidant capacity in tablespoons with grape flour studied by Dordevic et al. 2021 attributed to.

Response:

“Noticeablye, the addition of teff flour significantly influences the content of polyphenols in the tested products. The addition of polyphenol-rich flour has an enriching effect on the final product, e.g. the addition of grape flour increases the level of polyphenols and anti-oxidant properties, but the heating process at temperatures above 180 °C reduces poly-phenols and antioxidant properties even above 50% [56].”

  1. Regarding the biodegradability assay, with no intention of disregarding the obtained results, which are relevant considering the end of life of the studied material and product. In accordance with the test conditions, it can be stated that the materials are degradable under burial, yet following current standards, as established by the European standards EN 13432 / EN 14995 testing for certification include:

-       Chemical test: Disclosure of all constituents, threshold values for heavy metals are to be adhered to.

-       Biodegradability in controlled composting conditions (oxygen consumption and production of CO2): Proof must be made that at least 90 percent of the organic material is converted into CO2 within 6 months.

-       Disintegration: After 3 months composting and subsequent sifting through a 2 mm sieve, no more than 10 percent residue may remain, as compared to the original mass.

-       Ecotoxicity test: Examination of the effect of resultant compost on plant growth (agronomic test).

Response:

Thank you for your attention. We added the results from the following weeks that we received from the experience.

  1. Lastly, considering the reported phytotoxicity of olive pomace residues due to its high polyphenols content (Hachicha et al. 2009, Caporaso et al. 2017, Nunes et al. 2019). However as reported by Hachicha et al. 2009, the ecotoxicity of this by-product can be eliminated by co-composting with sesame bark. Given the polyphenols content of the materials reported in this work and the composition of the composite materials. May the carbon-rich compounds of the composite materials help reduce the ecotoxicity of the material during composting process? Do the authors believe an ecotoxicity test would be required?

Salma Hachicha, Juan Cegarra, Fatma Sellami, Ridha Hachicha, Noureddine Drira, K. Medhioub, Emna Ammar, Elimination of polyphenols toxicity from olive mill wastewater sludge by its co-composting with sesame bark. Journal of Hazardous Materials 161, 1131–1139 (2009). https://doi.org/10.1016/j.jhazmat.2008.04.066

  1. Caporaso, D. Formisano, A. Genovese, Use of phenolic compounds from. olive mill wastewater as valuable ingredients for functional foods. Critical Reviews in Food Science and Nutrition 58 (16), 2829-2841 (2018). http://dx.doi.org/10.1080/10408398.2017.1343797

M.A. Nunes, S. Pawlowski, A.S.G. Costa, R.C. Alves, M.B.P.P. Oliveira, S. Velizarov, Valorization of olive pomace by a green integrated approach applying sustainable extraction and membrane-assisted concentration.

Science of the Total Environment 652, 40–47 (2019). https://doi.org/10.1016/j.scitotenv.2018.10.204

Response:

We agree that the problem of phytotoxicity during biodegradability has been reported, however, as the reviewer mentioned, it depends on the substrate, but also on the pomace used. Our pomace is heated which changes its chemical composition compared to fresh pomace which we biodegrade. By looking at the literature recommended by the reviews, we will check the ecotoxicity of our products. Although we hope our packaging will be mainly consumed in the future.

Reviewer 2 Report

this study developed a biodegradable packaging from the olive oil wastes. I think this study is valuable when we consider sustainability, climate change and global warming problems. There is need for innovative and green packaging alternatives. 

-Please highlight the importance of the topic in the introduction part clearly

-Please state the needed references in the material-method section

-Please compare your result with similar studies, i couldnt see  discussion section

-I attached a pdf document please perform related  corrections 

-Please carefully check the language and correct some wrong words, please see line 378 "cheeses"???

-Please perform native english check.

Author Response

Dear Reviewer,

Thank you for your letter and constructive comments concerning our manuscript entitled “Quality assessment of waste from olive oil production and de-sign of biodegradable packaging”. We have read your comments carefully and made correction according to your suggestions to our manuscript. The most important changes are marked in green.

Thank you very much for your effort.

In the following, we give a point-by-point reply to your comments:

1.-Please highlight the importance of the topic in the introduction part clearly

Response:

Thanks for your suggestion. We have added important information emphasizing the importance of the topic.

2.-Please state the needed references in the material-method section

Response:

-We added literature.

-The DSC method was performed according to our knowledge, samples were tested at different temperatures and the best methodology was selected.

3.-Please compare your result with similar studies, i couldnt see  discussion section

Response:

Thank you for your comment. We added more research to the discussion.

4.-I attached a pdf document please perform related  corrections

Response:

Thanks again for a very good suggestion. The indicated errors in pdf are also shown in the sub-sections. We refer to those that have not been previously discussed.

-"edible" is a valid word that is commonly used in food

-Information about the patent application is in sex. 5. But we also added to the text. This is our own idea.

5.-Please carefully check the language and correct some wrong words, please see line 378 "cheeses"???

Response:

Corrected from “cheeses “ on “characteristics”. The manuscript has been checked and corrected by a native speaker.

6.-Please perform native english check.

Response:

The manuscript has been checked and corrected by a native speaker.

Reviewer 3 Report

Dear Authors, I belive that your research article wiil be further developedin line with the suggestions.

This manuscript written by Joanna Grzelczykal. is a valuable research on gastroparesis disease. " Quality assessment of waste from olive oil production and de- 2 sign of biodegradable packaging”. Although the subject is quite important, given the lack of further comparison of the study's findings with similar research on this topic in the literature, major revisions are required beforehand. Below I list some of the main issues that need to be improved.

-The explanations below need literature and research findings.

“Therefore, new packaging materials are being extensively explored, and  other agricultural products such as bran and corn are recently in use . However, these are  products of daily necessity, and there is increasingly insufficient food in the world. For  this reason, the production of new biodegradable products is highly profitable” (48-50).

-There is insufficient information in the Chemicals and Reagents heading. In this study, antioxidant activity analyzes were performed using the Dpph method. Solvent and chemicals used for this analysis need to be specified.  Many chemicals such as Folin-Ciocalteu, Na2CO3… were used. These are incomplete and need to add in the material section.

-There is a need for literature to support the confirmation of the comment made below in the DSC analysis results.

“The endothermic peak disappeared, which may indicate a lower water content and 238 greater thermal resistance, as the absence of an endothermic file in the 75-82 °C range may 239 mean that there is no non-surface-bound water in the baked packages, and may also indi- 240 cate a lack of retrograde starch”.

-The fact that the use of wax does not cause any change in L*a*b* values should be explained further, in fact, it should be interpreted in the direction of differences in the expected color change, or its relationship with the amount used.

-The use of cold or hot water is specified in the absorption process. The effect of using cold or hot water on absorption should be discussed. This must be a very important difference.

-According to the data in Table 6, there are significant increases in antioxidant activity at week 8, but this difference was not observed in polyphenol contents. Why do the two results not support each other? The factor that prevents the presence of polyphenols from directly affecting the antioxidant activity should be discussed.

-In the introduction section, the resistance of biodegradable packaging to microorganisms should be written as literature and research findings. The findings of the relevant studies are quite inadequate in this section. At the same time,  please add some references and examples that support this research topic. It will be more useful to benefit from these articles.

-A very high rate of plagiarism was observed in the explanations of antioxidant and polyphenol analyses. This needs improvement. I strongly recommend that you reduce this rate of plagiarism by referring some literature.

-Minor grammatical errors in English need to be corrected.

Author Response

Dear Reviewer,

Thank you for your letter and constructive comments concerning our manuscript entitled “Quality assessment of waste from olive oil production and de-sign of biodegradable packaging”. We have read your comments carefully and made correction according to your suggestions to our manuscript. The most important changes are marked in green.

Thank you very much for your effort.

In the following, we give a point-by-point reply to your comments:

1.-The explanations below need literature and research findings.

“Therefore, new packaging materials are being extensively explored, and  other agricultural products such as bran and corn are recently in use . However, these are  products of daily necessity, and there is increasingly insufficient food in the world. For  this reason, the production of new biodegradable products is highly profitable” (48-50).

Response:

We have added literature and research findings:

“However, these are products of daily necessity to produce of food, e.g., bread, and there is increasingly insufficient food in the world to which 1 in 9 people do not have access [31]. Climate change, which causes floods or droughts, also affects the growing deficit of agricultural products [32]. ”

2.-There is insufficient information in the Chemicals and Reagents heading. In this study, antioxidant activity analyzes were performed using the Dpph method. Solvent and chemicals used for this analysis need to be specified.  Many chemicals such as Folin-Ciocalteu, Na2CO3… were used. These are incomplete and need to add in the material section.

Response:

Thank you very much for this suggestion. Reagents have been added.

3.-There is a need for literature to support the confirmation of the comment made below in the DSC analysis results.

“The endothermic peak disappeared, which may indicate a lower water content and 238 greater thermal resistance, as the absence of an endothermic file in the 75-82 °C range may 239 mean that there is no non-surface-bound water in the baked packages, and may also indi- 240 cate a lack of retrograde starch”.

Response:

We added literature.

 “The endothermic peak disappeared, which may indicate a lower water content and great-er thermal resistance, as the absence of an endothermic file in the 75-82 °C range may mean that there is no non-surface-bound water in the baked packages and may also indi-cate a lack of retrograde starch [45-47]. According to Rolandella's research, the glass tran-sition temperature [45] increases with the increase in the water content in the product. Low water content and lower Tg indicate that the baked product may be more stable un-der storage conditions.

  1. -The fact that the use of wax does not cause any change in L*a*b* values should be explained further, in fact, it should be interpreted in the direction of differences in the expected color change, or its relationship with the amount used.

Response:

Thank you very much for your comments. We added an explanation.

“. The addition of beeswax changes the color only in large amounts. As the tested samples were immersed in the wax and then the wax surface was evened out with a brush, its thickness was so small that it did not brighten the final product. Disposable dishes were dark after baking, which also resulted in a lack of visual changes to the product. Beeswax does not have a strong, permanent dye in it [38,39].

  1. -The use of cold or hot water is specified in the absorption process. The effect of using cold or hot water on absorption should be discussed. This must be a very important difference.

Response:

Thanks again for a very good suggestion. As recommended, we have added a discussion of this topic.

“The water absorption factor is very important, as a general rule, once the temperature in-creases, the water is being absorbed faster [50]. Teff flour is characterized by higher water absorbability and gelling properties, which correlates with the results of DSC before bak-ing packaging [51]. The gelling properties allowed to plasticize the mass and make it eas-ier to form the vessels. Teff also promotes oil absorption, therefore it can be assumed that when combining the mixture with olive waste, it can absorb fat residues and consequently emulsify more strongly [52]. On the other hand, sorghum groats are characterized by the content of caffeine proteins, which is highly hydrophobic. The heat treatment of sorghum grits reduces its water activity, while increasing water absorption.  [53,54]. This suggests that the combination of olive pomace and teff flour and/or sorghum, although initially in-creased water absorption and increased viscosity of the product, reduced the water ab-sorption properties of both cold and hot water after baking. The addition of a beeswax coating additionally lowers the value of moisture absorption and gives it hydrophobic properties, which allows to extend the time of keeping the liquid in the cups, even when using hot water [55].”

.

6.-According to the data in Table 6, there are significant increases in antioxidant activity at week 8, but this difference was not observed in polyphenol contents. Why do the two results not support each other? The factor that prevents the presence of polyphenols from directly affecting the antioxidant activity should be discussed.

Response:

As for week 8, we disagree that there was no correlation in the results. Only in one case is the content of TPC (OTB + S) higher, in other tested samples it is lower than the initial content after 8 weeks. The tested DPPH value shows that we need more product to sweep away free radicals than in the case of a freshly baked cup. However, I understand that we need to discuss this relationship in more detail and we have added a description:

“The TPC content depends on many factors, it is not an exact method because it also shows other components than just polyphenols therefore, it increased the TPC content when storing products with a beeswax coating. The increase in TPC could also be influenced by the decomposition of polyphenols contained in beeswax [34,55,59].”

7.-In the introduction section, the resistance of biodegradable packaging to microorganisms should be written as literature and research findings. The findings of the relevant studies are quite inadequate in this section. At the same time,  please add some references and examples that support this research topic. It will be more useful to benefit from these articles.

Response:

In the literature, we can find mainly the microbiological evaluation of the film. We did not find microbiological tests on similar products as ours. However, we have added a description to the conclusions about the microorganisms that can be found in olive pomace.

“Populations of yeasts such as Candida, Pichia and fungi can be present in fresh olive pomace. However, this is dependent on the variety of the olives as well as the way the olive oil is produced, as it is important to use high-quality raw materials [49]. The pomace is additionally subjected to high temperature, which degrades the bacterial microflora. Additionally, the low activity of water in the final product has a positive effect on the shelf-life of the product.”

8.-A very high rate of plagiarism was observed in the explanations of antioxidant and polyphenol analyses. This needs improvement. I strongly recommend that you reduce this rate of plagiarism by referring some literature.

Response:

Thank you very much for this valuable comment. As recommended, we have added relevant literature on this topic. Changes were highlighted in the text (page 11, lines 351-359  ).

  1. -Minor grammatical errors in English need to be corrected.

Response:

The manuscript has been checked and corrected by a native speaker.

Round 2

Reviewer 1 Report

Dear authors,

I appreciate your responses and clarifications on the inquires raised on your work. The manuscript is clearer and presents a richer discussion of the results. The inclusion of the latest results degradation under burial of the materials remarks the substantiable approach of the work. As I have remarked before the article deals with two key environmental problems for sustainable production: plastic pollution due to single-use tableware and the reduction and valorization of by-products of the food industry. In this work the authors designed and studied fully biobased materials proposed as edible and disposable tableware. As stated by the authors further research on digestibility, nutritional characterization, and public acceptance for consuming the product as food, as well as ecotoxicity of the biodegraded material is needed. In this regard, a life cycle assessment on both end-of-life scenarios for the biobased tableware would be interesting for future work.

The article falls into the aims and scopes of the journal, and I recommend its publication after a minor revision. Please consider the following comments:

1.     Although the introduction has been improved, it is not clear from the literature review provided by the authors among which materials are the authors comparing the GHG emissions in this statement: “The use of fresh agricultural produce results in higher emissions of carbon dioxide into the atmosphere [26,27].” I believe that the authors refer the comparison to bioplastics based on biomass derived from industry by-products or waste since no additional resources are implied in the raw materials production and therefore GHG emissions in the process are reduced. Please revise this paragraph.

2.     Regarding DSC assays, the work by Rolandelli et al. (2022) cited indicates a decrease in Tg when water content of the material increase. Such effect is attributed to water plasticization effect, which increases mobility within the biopolymer matrix and other structural relaxation effects reported. If the product is storage below its Tg it will be more stable as reported by the authors, Therefore, lines 257-260 should be revised in this matter.

Author Response

We appreciate you for your precious time in reviewing our paper andmproviding valuable comments. It was your valuable and insightful comments that led to possible improvements in the current version.  We have carefully considered the  comments and tried our best to address every one of them. We hope the manuscript after careful revisions meet your high standards. All modifications in the manuscript have been highlighted in green.

1, 1.     Although the introduction has been improved, it is not clear from the literature review provided by the authors among which materials are the authors comparing the GHG emissions in this statement: “The use of fresh agricultural produce results in higher emissions of carbon dioxide into the atmosphere [26,27].” I believe that the authors refer the comparison to bioplastics based on biomass derived from industry by-products or waste since no additional resources are implied in the raw materials production and therefore GHG emissions in the process are reduced. Please revise this paragraph.

Response:
We agree with this comment and the revised version of the manuscript was complemented by this information. 

"The bioplastics based on biomass sources derived from industry by-products or waste reduce the emissions of GHG, since no additional resources are implied in the raw materials production [26,27]. ".

2.2.     Regarding DSC assays, the work by Rolandelli et al. (2022) cited indicates a decrease in Tg when water content of the material increase. Such effect is attributed to water plasticization effect, which increases mobility within the biopolymer matrix and other structural relaxation effects reported. If the product is storage below its Tg it will be more stable as reported by the authors, Therefore, lines 257-260 should be revised in this matter.

Response:

Thank you for your valuable comment. 

"According to Rolandella's research, the glass transition temperature [45] decrease with the increase in the water content in the product. It effect is attributed to enthalpy relaxations or structural relaxations found in biopolymer matrix. That the baked product may be more stable under storage conditions, should be stored at temperatures below their Tg."

Reviewer 3 Report

Dear Authors, I am pleased to accept your revised manuscript.

Author Response

Thank you for accepting the manuscript.